# Modeling and Analysis of N-Branch Hybrid Switched Inductor and Capacitor Converter

**Lei Yang [1],\*** , **Li Ma [1]**, **Xiaojie Li [1]**, **Liansong Xiong [2,3]**, **Xinghua Liu [1]** , **Hui Cao [3]** **and Junkang Ni [4]**

1   School of Electrical Engineering, Xi'an University of Technology, Xi'an 710048, China;
    2180321214@stu.xaut.edu.cn (L.M.); 2201920045@stu.xaut.edu.cn (X.L.); liuxh@xaut.edu.cn (X.L.)
2   School of Automation, Nanjing Institute of Technology, Nanjing 211167, China; xiongliansong@stu.xjtu.edu.cn
3   School of Electrical Engineering, Xi'an Jiaotong University, Xi'an 710049, China; huicao@mail.xjtu.edu.cn
4   School of Automation, Northwestern Polytechnical University, Xi'an 710072, China; junkangni@nwpu.edu.cn
\*   Correspondence: yanglei0930@xaut.edu.cn

**Abstract:** This paper proposes a family of N-Branch hybrid switched inductor and capacitor (SLC) converters. With the single circuit, the multi-level output voltage or current could be generated. The proposed converter is suitable both for the voltage source and the current source. The same LC network is reused for different LC branches. The proposed converter is controlled by the phase shift control method with a time domain multiplexing concept. The N level circuit is operated with the same frequency. One cycle period is divided into N small time cycles for each branch. The phase shift for each branch is $360°/N$. The load voltage could be changed by modifying the duty cycle of the transistor. When the SLCs work in the resonant condition, the soft switching will be acquired. The power loss of transistors could be sharply reduced. In this paper, a 300 W SLC converter is constructed to verify the theoretical analysis and operation mechanism in the resonant condition and hard switching condition. With the experimental and simulated verification, the soft switching and the stable multi-level output voltage or current are achieved. The proposed SLC converter could be used for the multi-level voltage power supply system, such as the electric vehicle, the electric aircraft, autonomous underwater vehicles (AUVs) and a new energy generation system.

**Keywords:** hybrid switched inductor and capacitor (SLC) converter; time domain multiplexing concept; soft switching; multilevel output voltages

## 1. Introduction

There are different kinds of DC–DC converters. The mostly popular one is the inductor-based converter. It includes a buck converter [1], boost converter, buck-boost converter, Cuk converter and so on. They could provide the stable voltage and current supply for different requirements. A switched capacitor (SC) converter has been widely used for renewable energy applications [2–4], implant medical devices [5–7], portable electric devices [8,9], data centers [10,11], and electric vehicles [12,13]. It has advantages such as ease of on-chip integration, being lightweight, and high-power density. The voltage conversion ratio of SC converters could be easily expanded through cascaded multiple SC cell topologies [14–18]. However, without the magnetic components, the current spikes will appear in the charge loop and discharge loop of a capacitor [19]. On the other hand, the input current of an SC converter will pulsate, leading to high current ripples and electromagnetic interference (EMI) noises [20,21]. The resonant switched-capacitor (RSC) converter is proposed for high-power applications. The small value inductor is injected into the charge loop or the discharge loop to reduce the spikes of the charging current and the discharging current [22,23]. What is more, the soft switching is achieved, which reduces the power loss of the transistors. However, in order to maintain the resonant condition, the load voltage is regulated in a very narrow range, which is hard for high-power applications.

The switched-inductor converter is proposed based on the switched-capacitor converter for the current source application [24–27]. It is widely used for PV system applications [28,29]. It provides an alternative power conversion method. This power conversion method is suitable for the current source and the current based load, such as LEDs. However, the application range of a switched-inductor converter is limited.

The inductor energy storage in the switched-inductor converter could reduce the current spikes. The capacitor energy storage in the switched-capacitor converter could reduce the ripples of the voltage. If the capacitor and inductor are used together, the quality of the converter could be highly improved, and the application range could then be largely extended. What is more, this kind of converter is suitable for both the current source and voltage source.

In a system, there are several required voltage levels for different applications [30–32]. The complex topology, such as buck-boost topology and modular multilevel topology, could provide the power supply for the different output voltage requirements. However, this method is high cost, and the power density is not high enough.

This paper proposes a family of N-Branch hybrid switched inductor and capacitor (SLC) converters; the combination of the capacitor energy storage and the inductor energy storage could bring an optimization of the traditional switched-capacitor converter and switched-inductor converter. It is both suitable for the current source and the voltage source. Moreover, it is good for the resistive load and the current based load. The pulsating input current and the input voltage ripples could be reduced with the proposed SLC converter. This converter provides multi-level voltage for different applications. It could be widely used for DC–DC power conversion.

The rest of the paper is organized as follows: the operation of the proposed SLC converter is shown in Section 2. Section 3 provides the theory analysis of the SLC converter. The theory verification is presented in Section 4. The conclusions and discussions are drawn in Section 5.

## 2. Operation Analysis of Proposed SLC Converter

The topology of the N-Branch SLC converter is shown in Figure 1. This SLC converter is a multi-level buck converter. Each branch circuit is divided and connected by a transistor. Based on the time domain multiplexing concept, the phase range of each branch is $\frac{360^\circ}{N}$. When the transistor is turned on in the Xth branch, the Xth branch is connected to the power source. In each branch, the load voltage could also be alternated by changing the duty-cycle of the transistor. A parallel connected LC network at the primary side of the SLC converter consists of the inductor $L_R$ and the capacitor $C_R$. The pulsating input current could be reduced with the inductor $L_R$, and the input voltage ripples could be filtered by the capacitor $C_R$. In each branch circuit, there is a serial connected inductor and capacitor. If $L_R = L_1 = L_2 = L_3 = \ldots = L_N$ and $C_R = C_1 = C_2 = C_3 = \ldots = C_N$, the proposed SLC converter works in the resonant condition. The zero-voltage switching and the zero current switching is achieved. What is more, the SLC converter could work in the hard-switching condition to provide a wide regulated voltage range. This SLC converter could achieve the continuous load voltage and continuous load current with the duty-cycle control in each branch.

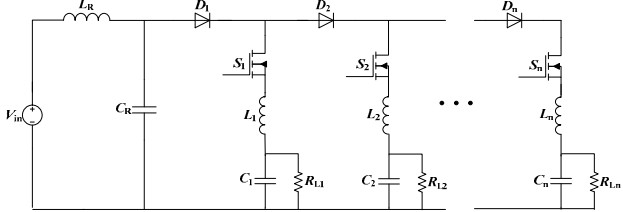

**Figure 1.** Topology of proposed N-Branch hybrid switched inductor and capacitor (SLC) converter.

The three-branch SLC converter is taken as an example to show the theory analysis and the operation method, which is shown in Figure 2. The timing waveforms of the

three-branch SLC converter are shown in Figure 3. It can be seen from Figure 3 that the phase shift of the transistor is set to $120°$. The one switching cycle could be divided into three stages. The period of Stage 1 is from $t_0$ to $t_2$. The period of Stage 2 is from $t_2$ to $t_4$ and the period of Stage 3 is from $t_4$ to $t_6$.

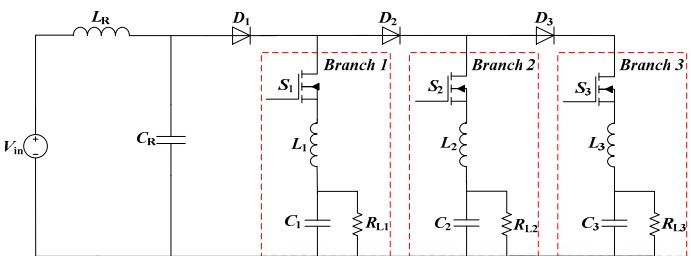

**Figure 2.** Three-branch SLC converter.

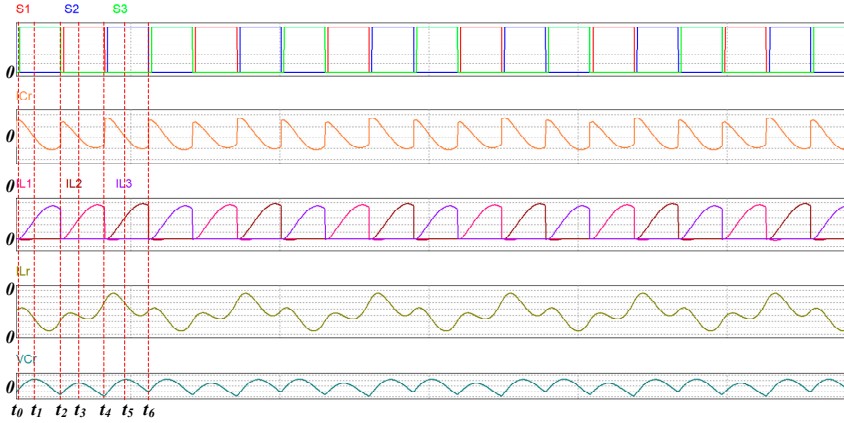

**Figure 3.** Timing waveforms of proposed SLC converter.

In Stage 1, $S_1$ is turned on while $S_2$ and $S_3$ are turned off; Branch 1 is connected to the power source. During the period ($t_0{\sim}t_1$), the inductor $L_R$ and capacitor $C_R$ are charged by the power source at the same time. The current of inductor $L_R$ and capacitor $C_R$ is in the forward direction. During the same period, the power source transfers its energy to the inductor $L_1$, capacitor $C_1$ and load resistor $R_{L1}$. During the period ($t_1{\sim}t_2$), the direction of current capacitor $C_R$ is changed. The power source, inductor $L_R$ and capacitor $C_R$ deliver their energy to inductor $L_1$, capacitor $C_1$ and load resistor $R_{L1}$.

In Stage 2, $S_2$ is turned on while $S_1$ and $S_3$ are turned off; Branch 2 is connected to the power source. The charge and the discharge of inductor $L_R$ and capacitor $C_R$ are similar to Stage 1.

In Stage 3, $S_3$ is turned on while $S_1$ and $S_2$ are turned off; Branch 3 is connected to the power source. The charge and the discharge of inductor $L_R$ and capacitor $C_R$ are similar to Stage 1.

The relative operating states of the proposed SLC converter are shown in Figure 4. The charge of capacitor $C_R$ and the discharge of capacitor $C_R$ are kept balanced in one switching cycle. Moreover, the charge of inductor $L_R$ and the discharge of inductor $L_R$ are also kept balanced in one switching cycle. The inductor $L_R$ and capacitor $C_R$ build a LC compensation network. It could work in the resonant condition to transfer the maximum power level for the three LC branches.

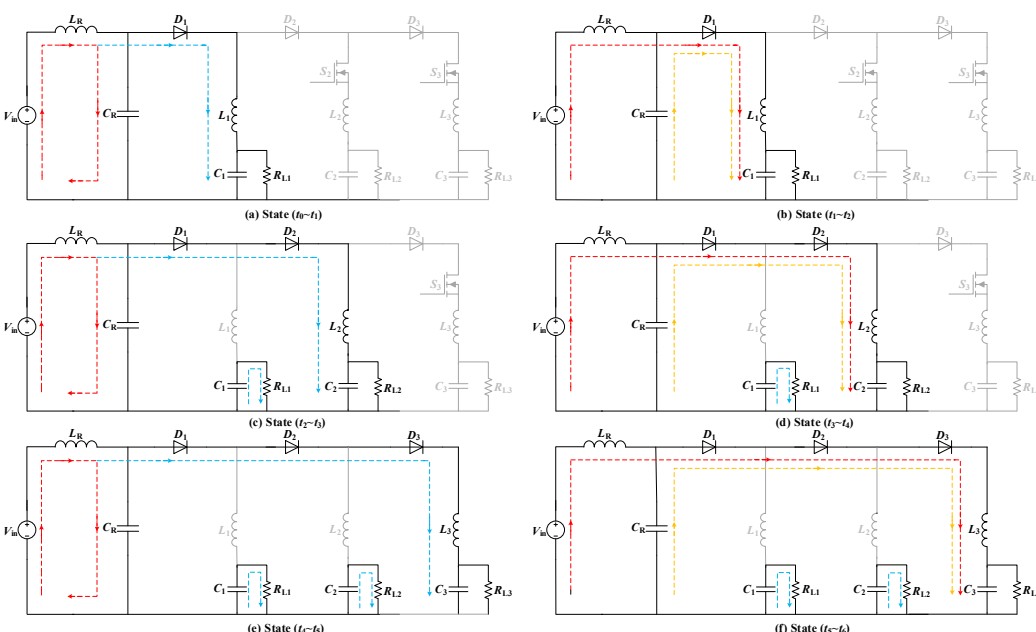

**Figure 4.** Operating states of proposed SLC converter.

The operation of the proposed SLC converter could be drawn in six states, as shown in Figure 4. During period ($t_0{\sim}t_2$), when $S_1$ is on while $S_2$ and $S_3$ are off, there are States 1 and 2. As shown in Figure 4a, in State 1, the input current flows to the capacitor $C_R$ and Branch 1. It can be seen from Figure 4b, in State 2, that the current of capacitor $C_R$ and the input current both flow to Branch 1. During period ($t_2{\sim}t_4$), when $S_2$ is on while $S_1$ and $S_3$ are off, there are States 3 and 4. As shown in Figure 4c, in State 3, the input current flows to the capacitor $C_R$ and Branch 2. It can be seen from Figure 4d, in State 4, that the current of capacitor $C_R$ and the input current both flow to Branch 2. During period ($t_4{\sim}t_6$), when $S_3$ is on while $S_1$ and $S_2$ are off, there are States 5 and 6. As shown in Figure 4e, in State 5, the input current flows to the capacitor $C_R$ and Branch 3. It can be seen from Figure 4f, in State 6, that the current of capacitor $C_R$ and the input current both flow to Branch 3.

## 3. Theory Analysis of Proposed SLC Converter

For easy understanding, in the calculation, the internal resistance of the power source and diodes is ignored. What is more, the voltages of the diodes are also neglected. For the effectual regulation of the load voltage, the duty-cycle of each branch should be set to less than $\frac{1}{3}$.

The resonant frequency of the primary compensation network could be written as:

$$f_s = \frac{1}{2\pi\sqrt{L_r C_r}} \tag{1}$$

where $L_r$ is the inductance of inductor $L_R$, and $C_r$ is the capacitance of capacitor $C_R$.

When $S_1$ is turned on while $S_2$ and $S_3$ are turned off, Branch 1 is connected to the power source. In Stage 1, during the period ($t_0{\sim}t_1$), as shown in Figure 4a, inductor $L_R$ and $C_R$ are charged by the power source. At the same time, the power delivers its energy to Branch 1. Based on the KVL principle, the following expressions are obtained:

$$L_r \frac{di_{Lr}(t)}{dt} = V_{in}(t) - V_{C1}(t) - V_{L1}(t) \tag{2}$$

$$i_{in}(t) = C_r \frac{dV_{Cr}(t)}{dt} + i_{in1}(t) \tag{3}$$

$$V_{Cr}(t) = V_{C1}(t) + V_{L1}(t) = V_{in} - V_{Lr}(t) \tag{4}$$

where $i_{\text{Lr}}(t)$ is the current of the inductor $L_{\text{R}}$, $V_{\text{in}}(t)$ is the input voltage, $V_{\text{C1}}(t)$ is the voltage of capacitor $C_1$, $V_{\text{L1}}(t)$ is the voltage of inductor $L_1$, $i_{\text{in}}(t)$ is the input current, $C_{\text{r}}$ is the capacitance of capacitor $C_{\text{R}}$, $V_{\text{Cr}}(t)$ is the voltage of capacitor $C_{\text{R}}$, and $i_{\text{in1}}(t)$ is the input current of Branch 1.

From Figure 3, the boundary condition is:

$$i_{\text{Cr}}(t_1) = 0 \tag{5}$$

During the period ($t_1 \sim t_2$), as shown in Figure 4b, the power source, inductor $L_{\text{R}}$, and capacitor $C_{\text{R}}$ deliver their energy to Branch 1 at the same time.

The following equations could be obtained:

$$i_{\text{in1}}(t) = C_{\text{r}}\frac{dV_{\text{Cr}}(t)}{dt} + i_{\text{in}}(t) \tag{6}$$

$$V_{\text{Cr}}(t) = V_{\text{in}}(t) + V_{\text{Lr}}(t) \tag{7}$$

When $S_2$ is turned on while $S_1$ and $S_3$ are turned off, Branch 2 is connected to the power source. In Stage 2, during the period ($t_2 \sim t_3$), as shown in Figure 4c, inductor $L_{\text{R}}$ and $C_{\text{R}}$ are charged by the power source. At the same time, the power delivers its energy to Branch 2. Based on the KVL principle, the following expression is obtained:

$$L_{\text{r}}\frac{di_{\text{Lr}}(t)}{dt} = V_{\text{in}}(t) - V_{\text{C2}}(t) - V_{\text{L2}}(t) \tag{8}$$

where $V_{\text{C2}}(t)$ is the voltage of capacitor $C_2$, and $V_{\text{L2}}(t)$ is the voltage of inductor $L_2$;

$$V_{\text{Cr}}(t) = V_{\text{in}}(t) - V_{\text{Lr}}(t) \tag{9}$$

$$i_{\text{in}}(t) = C_{\text{r}}\frac{dV_{\text{Cr}}(t)}{dt} + i_{\text{in2}}(t) \tag{10}$$

where $i_{\text{in2}}(t)$ is the input current of Branch 2.

It can be seen from Figure 3 that the boundary condition is:

$$i_{\text{Cr}}(t_3) = 0 \tag{11}$$

During the period ($t_3 \sim t_4$), as shown in Figure 4d, the power source, inductor $L_{\text{R}}$, and capacitor $C_{\text{R}}$ deliver their energy to Branch 2.

The following equations could be obtained:

$$i_{\text{in2}}(t) = C_{\text{r}}\frac{dV_{\text{Cr}}(t)}{dt} + i_{\text{in}}(t) \tag{12}$$

$$V_{\text{Cr}}(t) = V_{\text{in}}(t) + V_{\text{Lr}}(t) \tag{13}$$

It can be seen from Figure 3 that the boundary condition is:

$$i_{\text{Cr}}(t_4) = 0 \tag{14}$$

When $S_3$ is turned on while $S_1$ and $S_2$ are turned off, Branch 3 is connected to the power source. In Stage 3, during the period ($t_4 \sim t_5$), as shown in Figure 4e, inductor $L_{\text{R}}$ and $C_{\text{R}}$ are charged by the power source. At the same time, the power delivers its energy to Branch 3. Based on the KVL principle, the following expression is obtained:

$$L_{\text{r}}\frac{di_{\text{Lr}}(t)}{dt} = V_{\text{in}}(t) - V_{\text{C3}}(t) - V_{\text{L3}}(t) \tag{15}$$

where $V_{\text{C3}}(t)$ is the voltage of capacitor $C_3$, and $V_{\text{L3}}(t)$ is the voltage of inductor $L_2$;

$$V_{Cr}(t) = V_{in} - V_{Lr}(t) \tag{16}$$

$$i_{in}(t) = C_r \frac{dV_{Cr}(t)}{dt} + i_{in3}(t) \tag{17}$$

where $i_{in3}(t)$ is the input current of Branch 3.

It can be seen from Figure 3 that the boundary condition is:

$$i_{Cr}(t_5) = 0 \tag{18}$$

During the period ($t_5 \sim t_6$), as shown in Figure 4f, the power source, inductor $L_R$, and capacitor $C_R$ deliver their energy to Branch 3.

The following equations could be obtained:

$$i_{in3}(t) = C_r \frac{dV_{Cr}(t)}{dt} + i_{in}(t) \tag{19}$$

$$V_{Cr}(t) = V_{in}(t) + V_{Lr}(t) \tag{20}$$

It can be seen from Figure 3 that the boundary condition is:

$$i_{Cr}(t_6) = 0 \tag{21}$$

In one switching cycle, the charge and discharge of inductor $L_R$ and capacitor $C_R$ are kept balanced. Based on the aforementioned discussion and calculation, it can be seen from Figure 3 that the total charge of capacitor $C_R$ and total discharge of capacitor $C_R$ could be respectively derived as:

$$Q_{Cr(charge)} = \int_{t_0}^{t_1} C_r \frac{dV_{Cr}(t)}{dt} dt + \int_{t_2}^{t_3} C_r \frac{dV_{Cr}(t)}{dt} dt + \int_{t_4}^{t_5} C_r \frac{dV_{Cr}(t)}{dt} dt \tag{22}$$

$$Q_{Cr(discharge)} = [\int_{t_1}^{t_2} C_r \frac{dV_{Cr}(t)}{dt} dt + \int_{t_3}^{t_4} C_r \frac{dV_{Cr}(t)}{dt} dt + \int_{t_5}^{t_6} C_r \frac{dV_{Cr}(t)}{dt} dt] \tag{23}$$

In one switching cycle period, the charge balance of capacitor $C_R$ could be written as:

$$Q_{Cr(discharge)} + Q_{Cr(discharge)} = 0 \tag{24}$$

Substituting (22) and (23) into (24), the charge balance of capacitor $C_R$ in one switching cycle of capacitor could be derived as:

$$[\int_{t_0}^{t_1} C_r \frac{dV_{Cr}(t)}{dt} dt + \int_{t_2}^{t_3} C_r \frac{dV_{Cr}(t)}{dt} dt + \int_{t_4}^{t_5} C_r \frac{dV_{Cr}(t)}{dt} dt] + [\int_{t_1}^{t_2} C_r \frac{dV_{Cr}(t)}{dt} dt + \int_{t_3}^{t_4} C_r \frac{dV_{Cr}(t)}{dt} dt + \int_{t_5}^{t_6} C_r \frac{dV_{Cr}(t)}{dt} dt] = 0 \tag{25}$$

During the period ($t_0 \sim t_2$), the following equation does always stand up.

$$V_{Cr}(t) = V_{o1}(t) + V_{L1}(t) \tag{26}$$

During the period ($t_2 \sim t_4$), the following equation does always stand up.

$$V_{Cr}(t) = V_{o2}(t) + V_{L2}(t) \tag{27}$$

During the period ($t_4 \sim t_6$), the following equation could be derived.

$$V_{Cr}(t) = V_{o2}(t) + V_{L3}(t) \tag{28}$$

As a result, the average load voltages $V_{o1}$, $V_{o2}$ and $V_{o3}$ in one switching cycle could be expressed as:

$$V_{o1} = \frac{3 \int_{t_0}^{t_2} [V_{Cr}(t) - V_{L1}(t)] dt}{T_s} \tag{29}$$

$$V_{\text{o2}} = \frac{3 \int_{t_2}^{t_4} [V_{\text{Cr}}(t) - V_{\text{L2}}(t)] dt}{T_{\text{s}}} \tag{30}$$

$$V_{\text{o3}} = \frac{3 \int_{t_4}^{t_6} [V_{\text{Cr}}(t) - V_{\text{L3}}(t)] dt}{T_{\text{s}}} \tag{31}$$

In this paper, $D_1$ is set as the duty cycle of $S_1$, $D_2$ is set as the duty cycle of $S_2$ and $D_3$ is set as the duty cycle of $S_3$. The duty cycle could be changed to achieve the different load voltage.

If $D_1 \leq \frac{1}{3}$, $D_2 \leq \frac{1}{3}$, and $D_3 \leq \frac{1}{3}$, the average load voltages $V_{\text{o1}}$, $V_{\text{o2}}$, and $V_{\text{o3}}$, in one switching cycle, could be expressed as:

$$V_{\text{o1}} = \frac{3 \int_0^{D_1 T_{\text{s}}} [V_{\text{Cr}}(t) - V_{\text{L1}}(t)] dt}{T_{\text{s}}} \tag{32}$$

$$V_{\text{o2}} = \frac{3 \int_0^{D_2 T_{\text{s}}} [V_{\text{Cr}}(t) - V_{\text{L2}}(t)] dt}{T_{\text{s}}} \tag{33}$$

$$V_{\text{o3}} = \frac{3 \int_0^{D_3 T_{\text{s}}} [V_{\text{Cr}}(t) - V_{\text{L3}}(t)] dt}{T_{\text{s}}} \tag{34}$$

## 4. Experimental and Simulated Verification

In order to verify the performance and mechanism of the SLC converter, a 300 W power level SLC converter is constructed in this paper. The experimental SLC converter is shown in Figure 5. The power circuit and control unit are integrated in the same PCB board. The performance of the proposed SLC converter is respectively tested in the resonant condition and the normal condition. The parameters of the proposed SLC converter are shown in Table 1.

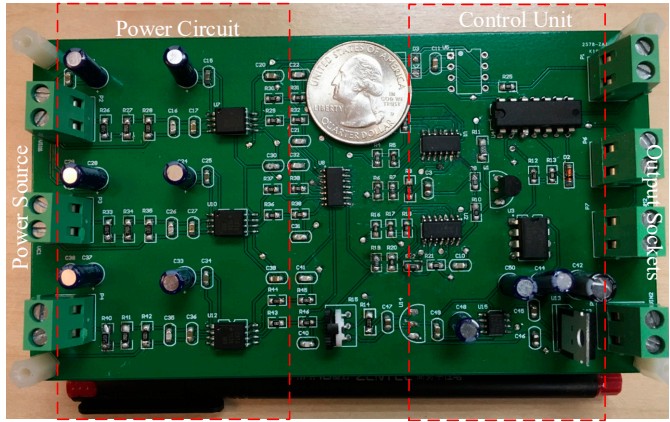

**Figure 5.** Experimental SLC converter with control unit.

**Table 1.** Circuit parameters.

| Prameters | Value |
|:---:|:---:|
| $L_{\text{r}}$ | 5 µH/20 µH |
| $C_{\text{r}}$ | 10 µF |
| $L_1, L_2, L_3$ | 5 µH |
| $C_1, C_2, C_3$ | 10 µF |
| $S_1, S_2, S_3, S_4$ | N channel MOSFET, IRF640NPBF, $V_{\text{dss}}$ = 200 V, $I_{\text{d}}$ = 18 A, 150 mΩ @ 11 A, 10 V, 150 W |

**Table 1.** *Cont.*

| Prameters | Value |
|---|---|
| $D_1, D_2, D_3$ | Fast Recovery Diode, VS-20CTH03STRL-M3, $V_r$ = 300 V, $I_o$ = 10 A, $V_f$ = 1.25 V@10 A, $t_{rr}$ = 35 ns. |
| Input voltage | 50 V and 100 V |
| Switching frequency | 22.51 kHz/100 kHz |
| $R_{L1} = R_{L2} = R_{L3}$ | 0~1000 Ω |

### 4.1. Case 1: Charge Balance Validation

As shown in Figure 6, if the switching frequency is set to 22.51 kHz, the proposed SLC converter works in the resonant condition. It can be seen that the waveforms of the input current and the currents of Branch 1, Branch 2 and Branch 3 meet the theory analysis. The smoothly input current is achieved. The current spikes of the transistors are highly reduced with the LC network in each branch. As shown in Figure 6, in one switching cycle, the summation of the charge of capacitor $C_R$ and the discharge of capacitor $C_R$ equal zero. This is the same for the inductor $L_R$. As a result, the charge balance of inductor $L_R$ and capacitor $C_R$ in one switching cycle is verified.

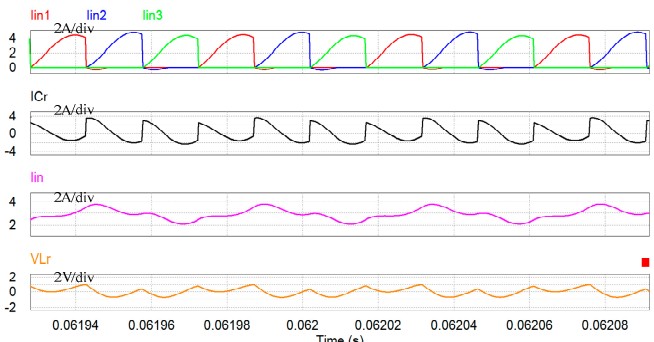

**Figure 6.** Operating waveforms of currents of proposed three-branch SLC converter.

### 4.2. Case 2: Soft Switching Validation

In the resonant condition, the ZVS and ZCS are achieved with the proposed SLC converter, as shown in Figure 7. When the switching frequency is set to 100 kHz (the duty cycle of $S_1$ is 0.2, the duty cycle of $S_2$ is 0.1 and the duty cycle of $S_3$ is 0.33), it can be seen from Figure 8 that the different voltages (33.19 V, 17.00 V and 41.23 V) are achieved by changing the duty-cycle of the transistor in each branch with the load resistance 50 Ω. The load voltage ripple factors of the three branches are respectively 0.80%, 0.85% and 0.69%. As a result, with a suitable regulation, the load voltage could be steadily controlled with good performance.

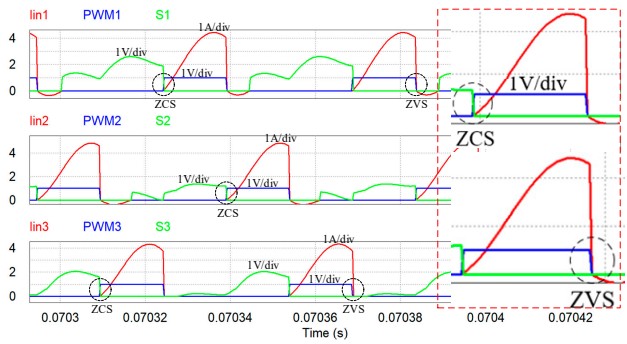

**Figure 7.** Soft switching of proposed three-branch SLC converter (voltage source).

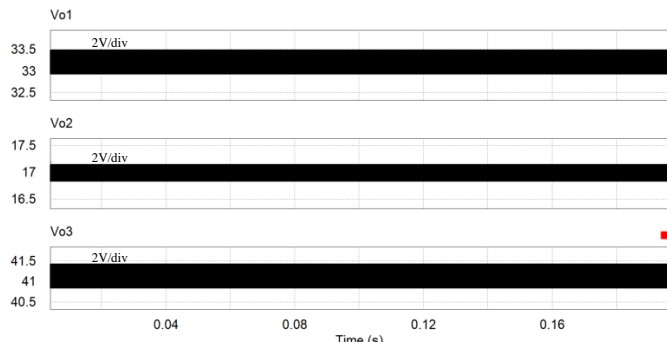

**Figure 8.** Operating waveforms of load voltages with frequency 100 kHz (voltage source).

### 4.3. Case 3: Current Source Validation

To demonstrate the suitable application for the current source, the simulation is conducted with a 5 A current source. When the proposed SLC converter works in the resonant condition, as can be seen in Figure 9, the input current is kept stable with very small ripples. The input currents of Branch 1, Branch 2 and Branch 3 are regulated smoothly without high spikes. The current spikes of the transistors are highly reduced with the LC network in each branch. In the resonant condition, as shown in Figure 10, the transistor could be turned on in the ZCS condition and turned off in the ZVS condition. The soft switching is achieved with the proposed SLC converter. It can be seen from Figure 11 that the load currents of the three branches are measured with the switching frequency of 100 kHz. When the switching frequency is set to 100 kHz (the duty-cycle of $S_1$ is 0.2, the duty cycle of $S_2$ is 0.1 and the duty cycle of $S_3$ is 0.33), as shown in Figure 11, the different currents (1.80 A, 0.93 A and 2.27 A) are achieved by changing the duty-cycle of the transistor in each branch with the load resistance 50 Ω. The load current ripple factors of the three branches are 0.79%, 0.87% and 0.69%. As a result, with a suitable regulation, the load current could be steadily controlled with good performance. The simulation results have a good match to the theory analysis.

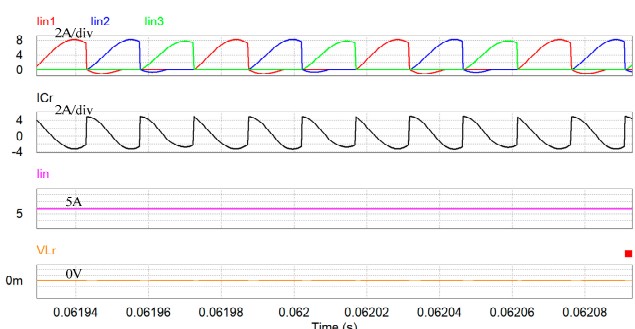

**Figure 9.** Operating waveforms of currents of proposed three-branch SLC converter (current source).

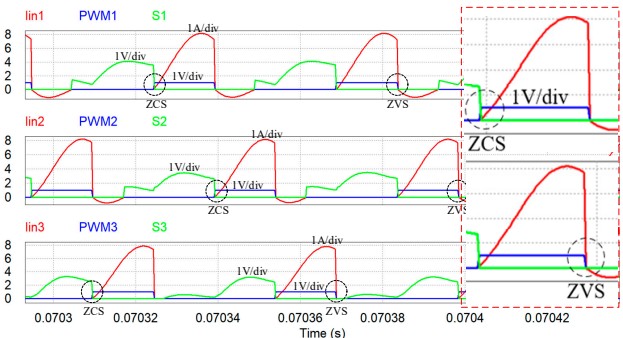

**Figure 10.** Soft switching of proposed three-branch SLC converter (current source).

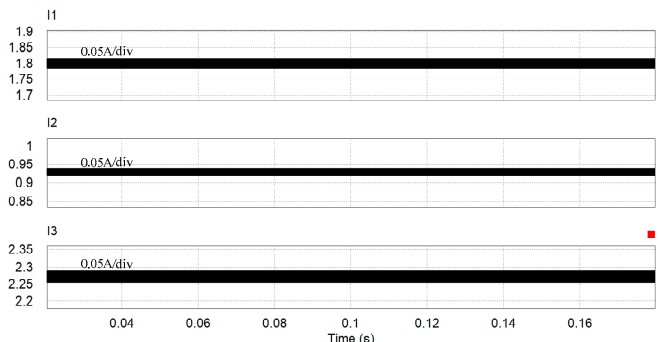

**Figure 11.** Operating waveforms of load voltages with frequency 100 kHz (current source).

As shown in Figure 12, with the same duty cycle of the transistors, the constant currents of the different branches will be achieved with different load power levels. It can be seen from Figure 12 that the input current source is set to 1 A, and the duty cycle of the three transistors of the three branches is set to 0.33. If the load resistors of Branch 1, Branch 2 and Branch 3 are set to 20 Ω, 50 Ω and 100 Ω, the load currents of the three branches are, respectively, 0.34 A, 0.33 A and 0.33 A. The constant current does not depend on the load power level. This is a specific feature of the SLC converter.

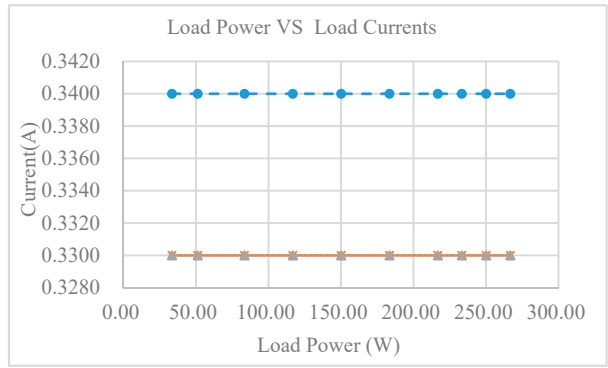

**Figure 12.** Operating waveforms of load currents with 1 A current source.

### 4.4. Case 4: Comparison of Voltage Source and Current Source

The operating waveforms of public capacitor $C_R$ and inductor $L_R$ are also measured in this paper. It can be seen from Figure 13 that, with the rising switching frequency, the voltage ripples of capacitor $C_R$ and the current ripples of inductor $L_R$ will be reduced. When the SLC converter works in the resonant condition (22.51 kHz), the sinusoidal voltage and sinusoidal current will be achieved with low EMI noise, as shown in Figure 13. With the 5 A current source, the value of the current ripples of inductor $L_R$ is almost zero. With the 100 V input voltage source, the maximum peak to peak voltage ripple of capacitor $C_R$ and the maximum peak to peak current ripple of inductor $L_R$ are respectively 3 V and 2 A.

The $V_o$ vs. duty-cycle and $I_o$ vs. duty-cycle of Branch 1 are respectively measured with the voltage source and the current source. The duty-cycle of $S_2$ and duty-cycle $S_3$ are both set to 0.31. The results are shown in Figure 14. The switching frequency is 100 kHz. It can be seen from Figure 14 that the output voltage and output current will both increase with the rising of the duty-cycle for the voltage source and the current source.

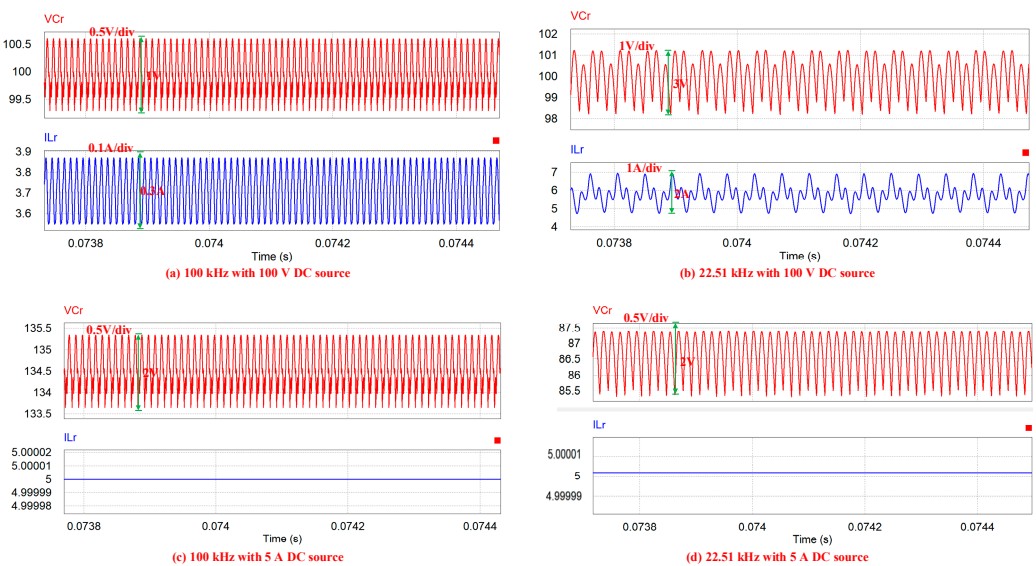

**Figure 13.** Operating waveforms of capacitor $C_R$ and inductor and inductor $L_R$.

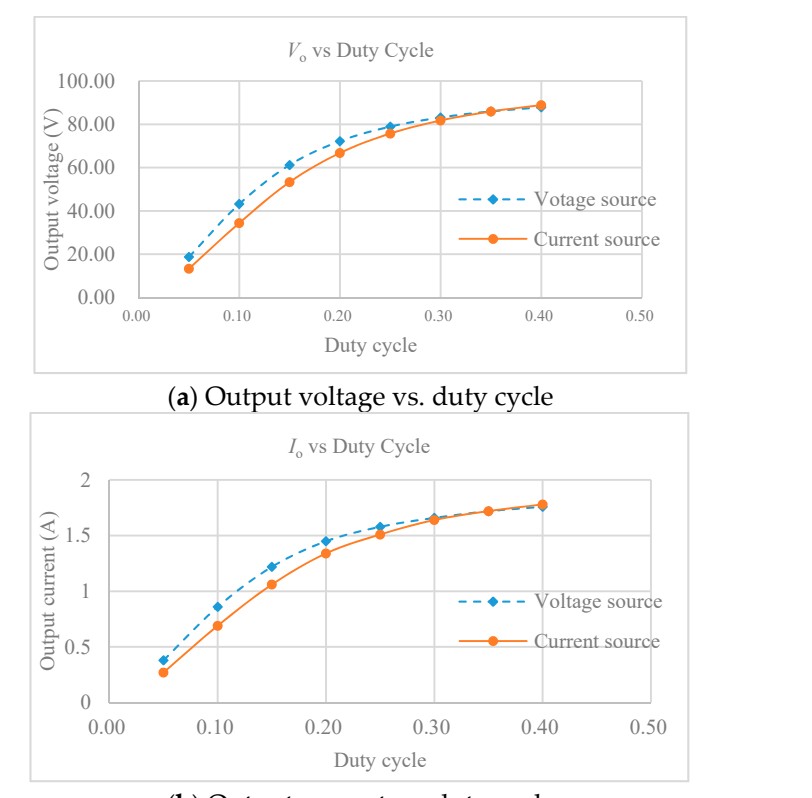

(**a**) Output voltage vs. duty cycle

(**b**) Output current vs. duty cycle

**Figure 14.** $V$o vs. duty-cycle and $I$o vs. duty-cycle of Branch 1.

*4.5. Case 5: Experimental Validation*

To verify the theory analysis, experiments were conducted in the laboratory. A 300 W power level three-branch SLC converter was built and tested as the experimental object. The voltage source was adapted in the experiments. The input voltages were respectively set to 50 V and 100 V. The duty-cycle of $S_1$ in branch 1 was set to 0.2. The switching frequency of all transistors was set to 100 kHz. As shown in Figure 15, the stable load voltage and current were achieved with 100 V input voltage. The input voltage of the three-branch SLC converter is continuous and smooth without high spikes. What is more,

when the input voltage was changed to 50 V and the duty-cycle was set to 0.3 for all transistors of the SLC converter, the tested waveforms of the load voltages of the three branches were obtained and are shown in Figure 16. It can be seen from Figure 16 that the output voltages of the three branches are similar. The amplitude is about 48 V. As a result, the multilevel output voltages could be acquired with the single SLC topology. The good performance of the SLC converter is verified with the simple topology.

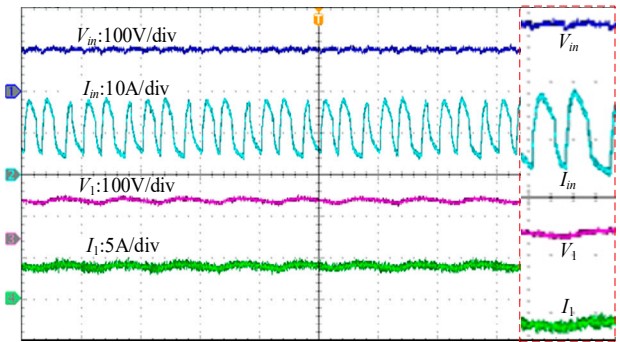

**Figure 15.** Load voltage and load current of Branch 1 with 100 V input voltage.

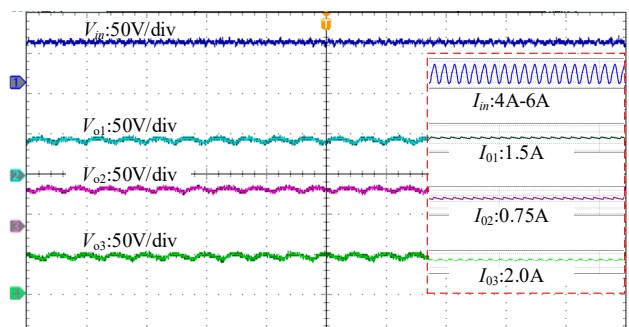

**Figure 16.** Load voltages and load currents of three-branch SLC converter with 50 V input voltage.

*4.6. Case 6: Efficiency Analysis*

The efficiency of the proposed SLC converter is respectively tested with the voltage source and the current source. The high efficiency is achieved with the proposed SLC converter. The tested results are respectively shown in Figures 17 and 18. The switching frequency is set to 100 kHz in the experiments. As shown in Figure 17, with the voltage source, the highest efficiency of the proposed SLC converter is 98.62% at the load power of 70 W. It can be seen from Figure 18 that, with the current source, the highest tested efficiency of the proposed SLC converter is 98.49% at the load power 270 W. The conclusion could be derived from the measured data, as shown in Figures 17 and 18, that, with the voltage source, the efficiency of the proposed SLC converter will decrease with the increasing of the load power; however, with the current source, the efficiency of the proposed SLC converter will climb with the increasing of the load power.

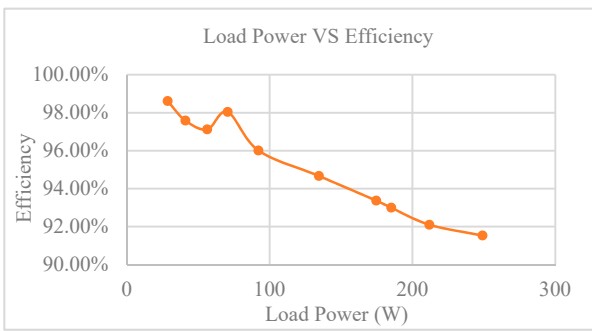

**Figure 17.** Measured efficiency of proposed SLC converter with the voltage source.

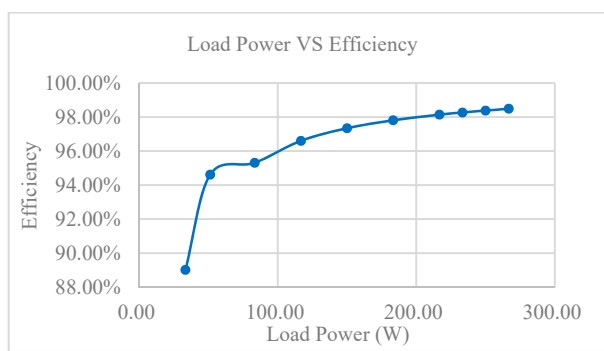

**Figure 18.** Measured efficiency of proposed SLC converter with the current source.

## 5. Conclusions and Discussion

An innovative N-Branch SLC converter is proposed in this paper. The LC resonant network is used for the power transfer unit. When the proposed SLC converter works in the resonant condition, the ZVS and ZCS are achieved. The proposed SLC converter is suitable for both the voltage source and the current source. The multi-level voltage or current could be realized by extending the branches and sharing the same primary LC network. The time domain multiplexing concept is used in this paper by dividing the one switching cycle phase to $\frac{360^{\circ}}{N}$. The PWM control method is adapted to get the required load voltage or current. Moreover, the continuous input current could be achieved with the voltage source. The simulated and experimental results show that the stable multi-level load voltage or current could be realized with the proposed SLC converter. The high efficiency of the proposed SLC converter is achieved with the simple topology. This SLC converter extends the application range of the converter for resistive loads and current based loads.

**Author Contributions:** L.Y. initiated the idea and designed the converter. L.M. and X.L. (Xiaojie Li) did the simulations and drafted the article. L.X., X.L. (Xinghua Liu), H.C. and J.N. initiated the conceptualization. L.M. did the formal analysis. All authors contributed towards extensive revisions of the article. All authors have read and agreed to the published version of the manuscript.

**Funding:** This work was supported by the China Postdoctoral Science Foundation under grant no.2018M643700, Scientific Research Project of Education Department of Shaanxi Province under grant no.18JS080, Postdoctoral Research Program of Shaanxi Province under grant no. 2018BSHY-DZZ28, Basic Research Project of Natural Science of Shaanxi Province under grant no.2020JQ-632, and Basic Research Project of Natural Science of Shaanxi Province under grant no.2020JQ-623.

**Conflicts of Interest:** The authors declare no conflict of interest. The funders had no role in the design of the study; in the collection, analyses, or interpretation of data; in the writing of the manuscript, or in the decision to publish the results.

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
