# Peer review of "Modeling and Analysis of N-Branch Hybrid Switched Inductor and Capacitor Converter"

_electronics, doi:10.3390/electronics10080891_

Round 1
Reviewer 1 Report
The research concept and design to propose a switched inductor-capacitor multi-level buck converter that can substantiate ZVS and ZCS seem interesting. Though the organization of the manuscript is quite good, authors are suggested to revise it thoroughly to correct a number of grammar and spelling mistakes. Also, the technical writing can be improved as well. The mathematical background to present state model analysis is good. However, there are certain prospects that are missing and required to be addressed in the revision, as follows.
a. The design of the SLC converter is not clearly presented. How are the transistors and diodes - which type of module, snubbers, power ratings of each switching device - chosen for the laboratory tests?
b. The capacitor voltage ripples and inductor current ripples (peak-to-peak for both) need to be presented.
c. Please check and modify Fig. 14, there is no current shown in this figure.
d. How is the converter operation in CCM or DCM mode controlled in the proposed topology? Generally, with the duty cycle parameter and observation of the inductor current ripples, the boundary condition between CCM and DCM modes are established.
e. More literature reviews on buck converter design and controls for real-time applications can be conducted and referred in this work, as suggested in the following.
S. Das, K. M. Salim and D. Chowdhury, "A novel variable width PWM switching based buck converter to control power factor correction phenomenon for an efficacious grid integrated electric vehicle battery charger," TENCON 2017 - 2017 IEEE Region 10 Conference, Penang, Malaysia.
f. The proposed model has been tested for 300 W, which is a reasonably low-level power application. However, it can not be claimed for universal high power applications, unless being tested for that scale.
Author Response
Response to Reviewer 1 Comments
Point 1: The research concept and design to propose a switched inductor-capacitor multi-level buck converter that can substantiate ZVS and ZCS seem interesting. Though the organization of the manuscript is quite good, authors are suggested to revise it thoroughly to correct a number of grammar and spelling mistakes. Also, the technical writing can be improved as well. The mathematical background to present state model analysis is good. However, there are certain prospects that are missing and required to be addressed in the revision, as follows.
Response 1: Thank you so much for your consideration and evaluation. We have revised according to your advice. The grammar and spelling mistakes have been corrected. The technical writing has also been improved. The certain prospects are addressed in the revised paper
Point 2: The design of the SLC converter is not clearly presented. How are the transistors and diodes- which type of module, snubbers, power ratings of each switching device - chosen for the laboratory tests?
Response 2: Thank you for your suggestions. The type of module, snubbers, power ratings of each switching device of transistors and diodes has been added as shown in Table 1in the revised paper in page 7.
Transistors:N channel MOSFET, IRF640NPBF, Vdss=200V, Id=18A, 150mΩ @ 11A,10V, 150W
Diodes:Fast Recovery Diode, VS-20CTH03STRL-M3, Vr=300 V, Io=10 A, Vf=1.25V@10 A, trr=35 ns.
Point 3: The capacitor voltage ripples and inductor current ripples (peak-to-peak for both) need to be presented.
Response 3:The capacitor voltage ripples and inductor current ripples (peak-to-peak for both) have been added in the revised paper.
The operating waveforms of public capacitor CR and inductor and inductor LR are shown in Fig.1(Figure 13 in the revised paper) that, with the rising switching frequency, the voltage ripples of capacitor CR and the current ripples of inductor LR will be reduced. When the SLC converter works in the resonant condition (22.51 kHz), the sinusoidal voltage and sinusoidal current will be achieved with low EMI noise as shown in Figure 1. With the current source, the value of current ripples of inductor LR is almost zero.
This revision has been highlighted in pages 9-10.
Point 4: Please check and modify Fig. 14, there is no current shown in this figure.
Response 4: The current waveforms have been added in Figure 2 (Figure15 in the revised paper) in the revised paper. This revision is shown in page 11 with yellow highlighter.
Point 5: How is the converter operation in CCM or DCM mode controlled in the proposed topology? Generally, with the duty cycle parameter and observation of the inductor current ripples, the boundary condition between CCM and DCM modes are established.
Response 5:The resonant frequency of the primary compensation network could be set according the responant switching equation. When the switching frequency is equal to the resonant frequency, the proposed converter works in the CCM condition. In this paper the resonant frequency is 22.51kHz.
Point 6: More literature reviews on buck converter design and controls for real-time applications can be conducted and referred in this work, as suggested in the following.
- Das, K. M. Salim and D. Chowdhury, "A novel variable width PWM switching based buck converter to control power factor correction phenomenon for an efficacious grid integrated electric vehicle battery charger," TENCON 2017 - 2017 IEEE Region 10 Conference, Penang, Malaysia.
Response 6:Thank you for your advice. More literature reviews on buck converter design and controls for real-time applications can be conducted and referred in this work. The paper with the title “A novel variable width PWM switching based buck converter to control power factor correction phenomenon for an efficacious grid integrated electric vehicle battery charger," has been cited in this paper as reference [1]. This revision has been highlighted in page 1.
Point 7: The proposed model has been tested for 300 W, which is a reasonably low-level power application. However, it can not be claimed for universal high power applications, unless being tested for that scale.
Response 7: Thank you for your suggestion. The statement of “universal converter” has been deleted in this paper. This revision has been red highlighted in page 12.

Reviewer 2 Report
Authors propose an N-branch hybrid switched inductor and capacitor converter, whose topology has been modeled and analyzed. A prototype with maximum output power of 300 W has been realized to validate the results of theoretical investigation.
The proposed system is interesting and paper is written enough well, however english proofreading is suggested.
Moreover, some suggestion are reported in the follow.
1) Abstract: authors use "theory analysis" instead of "theoretical analysis", it seems a typo. Along the paper, figures are referrenced as "Figure.$" with "$" number of figure. I think that the dot near "Figure" is unuseful.
2) Section 2: In Figure 3, the current I_Lr has positive and negative values, this is impossible since this current must be equal to the input current I_in for topological reasons. Can you explain that ?
3) Authors claim that during a a certain period the direction of the current i_Lr is changed, this is false. Why is this affermation reported ?
4) It could be better to report the voltage on the inductor Lr rather than the current flowing on it.
5) There is a typo in line 115. What means or what are charge/discharge electric qualities ?
6) Description of the working principle of the 3-stages converter during each single phase is recursive and written in the same manner, changing only the initial and final times of the conduction interval. Description could be made shorter and more compact.
7) As for the description, also theoretica analysis for each branch is ripetitive and can be avoided for the second and the third branch. Some equations are reported more the one time, it is preferred if a single equation is referred when occours. As an example, eqs. (3) and (7), or (9) and (13) and so on.
8) Voltage and current directions of the components should be reported at the least one subfigure of figure 4, to allow the reader a better comprehension of the carried out analysis.
9) I have some dubt with equation (6), KVL of the input electric mesh agrees with equation (4), which is opposed to eq. (6). It creates ununiformity between equations. I think that authors want to include the change of sign of the voltage across the inductor L_R on the equation. I advice to uniform the various equations and specify when the voltages or currents change their signs due to changes in topology and/or conditions.
10) Maybe there is a typo in line 290, Figure 5 is not rightly referred.
11) What are the maximum output currents and voltages in both cases of input current source and voltage source ? Please, report it.
12) It could be interesting if graphs depicting V_OUT vs. duty-cycle and I_OUT vs. duty-cycle of a single branch are added for both validation cases, input current and voltage sources.
13) Some referrences are missing: "Regulated charge pumps: A comparative study by means of verilog-AMS" and "A Review of Power Management Integrated Circuits for Ultrasound-Based Energy Harvesting in Implantable Medical Devices" for SC circuits for implanted medical devices.
"Linear distribution of capacitance in Dickson charge pumps to reduce rise time" and "Charge pump improvement for energy harvesting applications by node pre-charging" for SC circuits with cascade of SC cell topologies.
Author Response
Response to Reviewer 2 Comments
Point 1: Authors propose an N-branch hybrid switched inductor and capacitor converter, whose topology has been modeled and analyzed. A prototype with maximum output power of 300 W has been realized to validate the results of theoretical investigation.
The proposed system is interesting and paper is written enough well, however english proofreading is suggested.
Response 1: Thank you so much for your consideration and suggestions.
Point 2: Abstract: authors use "theory analysis" instead of "theoretical analysis", it seems a typo. Along the paper, figures are referrenced as "Figure.$" with "$" number of figure. I think that the dot near "Figure" is unuseful.
Response 2: Thank you for your advice. The "theory analysis" has been revised as "theoretical analysis" as shown in page 1 and the dot near "Figure" has been deleted in the revised paper.
Point 3: Section 2: In Figure 3, the current I_Lr has positive and negative values, this is impossible since this current must be equal to the input current I_in for topological reasons. Can you explain that ?
Response 3: Thank you for your advice. This mistake has been revised in the updated paper. The Figure 3 has been redrawn without negative current of inductor LR. This revision has been highlighted in page 3.
Point 4: Authors claim that during a certain period the direction of the current i_Lr is changed, this is false. Why is this affermation reported ?
Response 4: Thank you for your advice. This mistake has been revised. The Figure 3 has been redrawn, as shown in Figure 4 (Figure 3 in the revised paper), there is no negative current of inductor LR .
Point 5: It could be better to report the voltage on the inductor Lr rather than the current flowing on it.
Response 5: Thank you for your suggestion. The voltage waveforms of inductor Lr could be found in Figure 6 in page 7.
Point 6: There is a typo in line 115. What means or what are charge/discharge electric qualities ?
Response 6: Thank you for your question. The “charge/discharge electric qualities” has been revised as “charge/discharge” in the revised paper. This revision has been highlighted in page 3.
Point 7: Description of the working principle of the 3-stages converter during each single phase is recursive and written in the same manner, changing only the initial and final times of the conduction interval. Description could be made shorter and more compact.
Response 7: The working principle of the 3-stages converter during each single phase description is made shorter and more compact in the revised paper. The description of Stage 2 and Stage 3 is shortened as:
“In Stage 2, S2 is turned on while S1 and S3 are turned off, the Branch 2 is connected to the power source. The charge and the discharge of inductor LR and capacitor CR are similar to Stage 1.
In Stage 3, S3 is turned on while S1 and S2 are turned off, the Branch 3 is connected to the power source. The charge and the discharge of inductor LR and capacitor CR are similar to Stage 1.”
This revision is shown with highlighter in page 3.
Point 8: As for the description, also theoretica analysis for each branch is ripetitive and can be avoided for the second and the third branch. Some equations are reported more the one time, it is preferred if a single equation is referred when occours. As an example, eqs. (3) and (7), or (9) and (13) and so on.
Response 8: Thank you for your suggestion. The repeated equations have been deleted in the revised paper.
Point 9: Voltage and current directions of the components should be reported at the least one subfigure of figure 4, to allow the reader a better comprehension of the carried out analysis.
Response 9: The Voltage and current directions of the components has been added in the revised paper.This revision is highlighted in page 3 in the revised paper.
Point 10: I have some dubt with equation (6), KVL of the input electric mesh agrees with equation (4), which is opposed to eq. (6). It creates ununiformity between equations. I think that authors want to include the change of sign of the voltage across the inductor L_R on the equation. I advice to uniform the various equations and specify when the voltages or currents change their signs due to changes in topology and/or conditions.
Response 10: After discussion, we have changed the description. We think the eq.(6) is not accurate. As a result, the eq. (6) has been deleted in the revised paper.
Point 11: Maybe there is a typo in line 290, Figure 5 is not rightly referred.
Response 11: The Figure 5 in the sentence “As shown in Figure.5, in one switching cycle, the summation of the charge electric quality of capacitor” has been revised as “Figure 6”. This revision has been highlighted in page 7.
Point 12: What are the maximum output currents and voltages in both cases of input current source and voltage source ? Please, report it.
Response 12: Thank you for your question. With the 100V voltage source, the maximum output current and voltage are respectively 5 A and 95 V. With the 5 A current source, the maximum output current and voltage are respectively 1.7 A and 85 V.
Point 13: It could be interesting if graphs depicting V_OUT vs. duty-cycle and I_OUT vs. duty-cycle of a single branch are added for both validation cases, input current and voltage sources.
Response 13: Thank you for your advice. The graphs depicting V_OUT vs. duty-cycle and I_OUT vs. duty-cycle of a single branch are added for both validation cases, input current and voltage sources have been added in the revised paper. It is shown as Figure 5 (Figure 14 in the revised paper) in page 10.
The Vo vs. duty-cycle and Io vs. duty-cycle of a Branch 1 are measured with the voltage source and the current source. The results are shown as Figure 14. The duty-cycle of S2 and S3 is set 0.31. The switching frequency is 100 kHz. It can be seen from Figure 14 that the output voltage and output current will increase with the rising of duty-cycle both for the voltage source and the current source.
Point 14: Some referrences are missing: "Regulated charge pumps: A comparative study by means of verilog-AMS" and "A Review of Power Management Integrated Circuits for Ultrasound-Based Energy Harvesting in Implantable Medical Devices" for SC circuits for implanted medical devices.
"Linear distribution of capacitance in Dickson charge pumps to reduce rise time" and "Charge pump improvement for energy harvesting applications by node pre-charging" for SC circuits with cascade of SC cell topologies.
Response 14: Thank you for your advice, the references have been added in this paper as shown in page 13.

Reviewer 3 Report
Dear Authors,
the manuscript and the topic are interesting; the presentation quality is good and I have some minor comments.
1) Line 34. The two "the" are redundant.
2) Line 50. What you mean with "restricted"? It means "not allowed", but you mean maybe "limited".
3) Line 287-288. Please, rewrite the sentence for better English form. "It can be seen ... that ... input current and the currents .... meet ..."
4) Figure 7 and Figure 10. Please, could you provide a zoom of the two ZCS and ZVS areas, in order to appreciate the details of the waveforms during the transition?
5) Line 325. Could you explain what you men with switching frequency "adapted"?
6) Line 327-328. The sentence beginning with "what's more ..." is hanging.
7) Figure 13 and 14. First, label should be "Figure". Then, please, could you provide images with better resolution and possibly reduced time scale (e.g. half of the one presently used), so the Iin periods and ripple can be seen better?
Could you also add the detail of the time scale in the figures?
8) References. Please, provide references in MDPI style.
9) Line 340. Sentence "the high efficiency ..." is redundant and should be better worded ad at the beginning of the subsection.
10) Figure 15. Could you comment the local maximum at about 70W loading, giving a justification for the behavior?
11) Figure 15 ad 16. Could you comment why the efficiency curves have opposite behavior in voltage and current source conditions?
Author Response
Response to Reviewer 3 Comments
Point 1: Line 34. The two "the" are redundant.
Response 1: Thank you for your advice. The redundant “the” has been deleted in the revised paper. This revision has been highlighted with red color in page 1.
Point 2: Line 50. What you mean with "restricted"? It means "not allowed", but you mean maybe "limited".
Response 2: Thank you for your suggestion. The word “restricted” has been changed as “limited”. This revision is shown in page 2 with the red color.
Point 3: Line 287-288. Please, rewrite the sentence for better English form. "It can be seen ... that ... input current and the currents .... meet ..."
Response 3: Thank you for your advice. The sentence has been rewritten as “It can be seen that the waveforms of input current, the currents of Branch 1, Branch 2 and Branch 3 meet the theory analysis”. This revision could be seen in page 7 with yellow color.
Point 4: Figure 7 and Figure 10. Please, could you provide a zoom of the two ZCS and ZVS areas, in order to appreciate the details of the waveforms during the transition?
Response 4: Thank you for your good advice, the zoomed ZCS and ZVS areas are respectively added in Figure 6 (Figure 7 in the revised paper) and Figure 7 (Figure 10 in the revised paper) in the revised paper. Theses revisions have been highlighted in pages 8 and 9 .
Point 5: Line 325. Could you explain what you men with switching frequency "adapted"?
Response 5. Thank you for your suggestion. The switching frequency "adapted" means that the “The switching frequency of all transistors is set 100 kHz”. This sentence has been revised in page 11 with the yellow highlighter.
Point 6: Line 327-328. The sentence beginning with "what's more ..." is hanging.
Response 6: Thank you for your good catch. The sentence “what's more ...” has been changed as “What’s more, when the input voltage is changed to 50 V and the duty-cycle is set 0.3 for all transistors of SLC converter, the tested waveforms of load voltages of three branches are shown as Figure.16.” This revision has been highlighted with red color in page 11 in the revised paper.
Point 7: Figure 13 and 14. First, label should be "Figure". Then, please, could you provide images with better resolution and possibly reduced time scale (e.g. half of the one presently used), so the Iin periods and ripple can be seen better?
Could you also add the detail of the time scale in the figures?
Response 7: The label of Figure 13 (Figure 15 in the revised paper) and Figure 14 (Figure 16 in the revised paper) has been revised. The detail of the time scale in the figures has been added in the revised paper. This revision is shown in page 11.
Point 8: References. Please, provide references in MDPI style.
Response 8: Thank you for your advice. The references have been revised in MDPI style.
Point 9: Line 340. Sentence "the high efficiency ..." is redundant and should be better worded ad at the beginning of the subsection.
Response 9: Thank you for your good advice. The sentence “The high efficiency is achieved with the proposed SLC converter ” has been placed at the beginning of the subsection. This revision has been highlighted in page 11.
Point 10: Figure 15. Could you comment the local maximum at about 70W loading, giving a justification for the behavior?
Response 10: Thank you for your suggestion. The power loss will be rising with the high current ripples. As a result, the efficiency will decrease.
Point 11: Figure 15 ad 16. Could you comment why the efficiency curves have opposite behavior in voltage and current source conditions?
Response 11: Thank you for advice. In order to explain this behavior, we conducted more research work. As shown in Figure 10 (Figure 13 in the revised paper), with the voltage source, the higher current ripples of inductor LR will occur. The SLC converter will generate more power loss. As a result, the efficiency of SLC converter will have a decreasing trend with the rising power level. However, with the current source, the current ripples of inductor LR is almost zero. As a result, the efficiency of SLC converter will have an increasing trend with the rising power level.

Round 2
Reviewer 1 Report
The revised manuscript addresses the recommendations quite fine.
Thank you.